# SurvivMIL: A Multimodal, Multiple Instance Learning Pipeline for Survival Outcome of Neuroblastoma Patients

**Reed Naidoo**                                 REED.NAIDOO@ICR.AC.UK
**Olga Fourkioti**                           OLGA.FOURKIOTI@ICR.AC.UK
**Matt De Vries**                               MATT.DEVRIES@ICR.AC.UK
**Chris Bakal**                                   CHRIS.BAKAL@ICR.AC.UK
*The Institute of Cancer Research, London, UK*

## Abstract

Integrating Whole Slide Images (WSIs) and patient-specific health records (PHRs) can facilitate survival analysis of high-risk neuroblastoma (NB) cancer patients. However, this integration is challenging due to extreme differences in data dimensionality. Specifically, while PHRs are at the patient level and contain sparse information, WSIs are highly information-dense and processed at high resolution. Adjacent to this challenge, specifically in the context of survival analysis under the Multiple Instance Learning (MIL) framework, there are limitations with approximating the hazard function because of varying size WSIs and implicitly limited batch sizes. To address these challenges, we propose SURVIVMIL, a late fusion MIL model that integrates multimodal prognostic data for predicting NB patient outcomes. Our approach fuses predictions from both modalities and incorporates a novel concordance-based loss function via a specifically designed buffer branch, which mitigates the batch size limitation by accumulating survival predictions. Our model is evaluated on an in-house pediatric NB patient dataset, providing insights into the contributions of each modality to predictive performance. The code will be available at: `https://github.com/reednaidoo/SurvivMIL_COMPAYL.git`

**Keywords:** Multiple Instance Learning, Multimodal Fusion, Digital Pathology, Survival Analysis

## 1 Introduction

The analysis of hematoxylin and eosin (H&E) stained Whole Slide Images (WSIs) has become indispensable in digital pathology, playing a pivotal role in extracting meaningful features for the precise diagnosis (Lu et al., 2021; Campanella et al., 2019; Wang et al., 2016) and treatment (Litjens et al., 2016; Yao et al., 2020; Pinckaers et al., 2020) of patients. Notably, we have witnessed the refined development of AI-based models geared towards the analysis of H&E stained WSIs, with techniques often surpassing expert pathologists' performance (Srinidhi et al., 2021; Wang et al., 2021; Das et al., 2018; Tong et al., 2014; Melendez et al., 2015; Quellec et al., 2016). Weakly supervised Multiple Instance Learning (MIL) classifiers have gained prominence as prevalent methodologies for handling this type of data, with innovative approaches now incorporating vision and graph transformers, along with considerations for tile relationships (Fourkioti et al., 2024; Shao et al., 2021; Zheng et al., 2022).

In addition to histological data, high-throughput technologies have drastically increased the volume of genomic, proteomic, and transcriptomic data available for cancer research. This 'omics' data offers a granular view of the molecular underpinnings of heterogeneous cancers, which are driven by numerous genomic alterations. Analysis of this data could sub-type patients, moving towards more precise diagnosis and prognosis in cancers such as neuroblastoma (NB).

NB is a prevalent pediatric malignancy, representing the most frequent cancer diagnosis within the first year of life and contributes to over 15% of pediatric cancer-related deaths (Watanabe et al., 2022). Presently, patients categorised as 'high-risk' receive uniform intensive clinical treatment, disregarding the inherent heterogeneity observed across high-risk patient profile modalities (Moreno et al., 2021). Creating a methodology capable of accurately classifying a subset of high-risk NB patients who are particularly prone to unfavourable outcomes holds the potential to enhance precision in diagnosis at an earlier stage. This could facilitate timely access to innovative therapies for these patients (Moreno et al., 2021). Presently, the International Neuroblastoma Risk Group Staging System (INRG) (Monclair et al., 2009) identifies age and MYCN amplification status of a patient as predictive biomarkers for NB patient outcomes. Leveraging histological data with these patient-specific biomarkers in NB survival outcome prediction could further refine patient stratification into risk subgroups.

Integrating such modalities under the framework of multimodal oncology often centres around data fusion strategies (Stahlschmidt et al., 2022). This aims to leverage complementary information from different modalities to improve decision-making and interpretability. These strategies involve the concatenation, element-wise sum, multiplication, or Kronecker product of various modalities at different stages, classifiable as early, intermediate, or late fusion (Huang et al., 2020). Because of the heterogeneous nature of biomedical data modalities, current early fusion strategies have investigated co-attention modules to model the attention of histology patches towards gene sets (Chen et al., 2021). Extending on cross-attention modelling that evaluated WSI patch-to-gene interactions, authors have also studied dual patch-to-pathway and pathway-to-patch interactions for effective survival analysis (Jaume et al., 2024).

These dense multimodal early fusion approaches, particularly those employing cross-attention techniques, have effectively integrated WSIs and dense omic modalities for patient risk stratification due to the closer data alignment between these two modalities. However, in our study, we encountered clinically relevant prognostic biomarkers in NB, specifically age and MYCN status, that exhibit extreme data dimensionality gaps compared to WSIs. This disparity complicates the integration of such diverse data types, making early fusion data alignment impractical in this setting. Consequently, there is limited research on integrating such disparate data types to enhance survival analysis in NB.

This paper proposes a late fusion multi-branch MIL architecture, dubbed SurvivMIL, that considers H&E-stained WSIs and patient-specific health record data, addressing the challenge of extreme multimodal dimensionality gaps in predicting NB patient outcomes. Additionally, we introduce a concordance-based loss function specifically designed for the survival outcome problem, which penalises the model for incorrect ranking of survival predictions. Our proposed architecture demonstrates improved model performance, and we

further evaluate different settings of modality contribution for joint prediction of patient outcomes.

## 2 Methods

Inspired by previous work (Jaume et al., 2024), we define a censorship status, $c$, and a time-to-event, $t$, to frame our supervised survival prediction task. The censorship status indicates whether we observed a patient death; $c = 1$ indicates a known follow-up, and $c = 0$ indicates an observed patient death. The time-to-event, $t$, represents the time from the biopsy to the date of death (if $c = 0$) or to the follow-up date (if $c = 1$). The supervised classification task is then defined by dividing the given time responses into non-overlapping quartiles, $y_j$: $[t_{j-1}, .., t_j)$ where $j \in [1, ..., n]$. Each output logit, $\hat{y}_j$ of the multiclass classifier corresponds to a specific time quartile. The hazard function is defined as:

$$\mathbf{h} = \sigma(\hat{y}_j), \tag{1}$$

where $\mathbf{h}$ is the probability of survival of a patient during the time interval $(t_{j-1}, t_j)$ and $\sigma$ is the sigmoid activation function. From the discrete hazard function, the predicted probability of survival is thus defined:

$$S = \prod_{k=1}^{j}(1 - \mathbf{h}_j) \tag{2}$$

To conduct survival analysis and effectively learn the hazard function, we propose an integrated architecture combining WSIs and PHRs. SURVIVMIL consists of a feature extraction pipeline followed by a multi-branch classifier. Within the classifier, we implement a concordance-based loss function. The details of these components are discussed in the following subsections.

### 2.1 WSI Bag Construction

We adopt the strategy of self-supervised contrastive learning, specifically the UNI model (Chen et al., 2024) to extract semantically rich, meaningful feature representations. The UNI model is based on DINOv2 (Oquab et al., 2024), a state-of-the-art self-supervised learning method based on student-teacher knowledge distillation for pretraining large Vision Transformer (ViT) architectures (Dosovitskiy et al., 2021). Specifically, this pre-trained feature extractor, $f(\cdot)$, is utilised to produce a set of features:

$$\mathbf{B}_k = \{\mathbf{i}_1, \mathbf{i}_j, ..., \mathbf{i}_{n_{B_k}}\}, \mathbf{i}_j \in R^{1 \times 1024}, \tag{3}$$

representing the bags of instance embeddings of the tiled WSIs.

### 2.2 Multi-branch classifier

We adopt a late fusion approach in our multi-branch classifier, combining the global outputs from the WSI data with those from PHRs. By merging these data branches at a higher level of abstraction, this approach ensures that both the fine-grained details captured by the WSIs and the broad PHRs contribute to the final prediction of patient outcomes.

Our model comprises two branches: a WSI branch and a PHR branch. The set of instance embeddings $i_j \in R^{1 \times 1024}$, produced by the feature extractor, are input into the WSI branch. Similar to the DSMIL architecture, the WSI branch employs both an instance-level and a bag-level classifier. The instance-level classifier performs a max-pooling operation to determine a critical instance. Subsequently, the bag-level classifier utilises a non-local attention mechanism to evaluate the importance of each instance and aggregates the instance embeddings to form a collective representation. The slide-level representation is a weighted average of the attention scores produced by the non-local attention mechanism. These scores, $\mathbf{a}_{B_k} = \{\mathbf{a}_{i1}, ..., \mathbf{a}_{in_{B_k}}\}$, are computed by measuring the similarity between the instance embeddings and the critical instance, enabling the model to focus on the most relevant instances for the classification task. These two streams are trained concurrently and merged at the end of the WSI classifier branch.

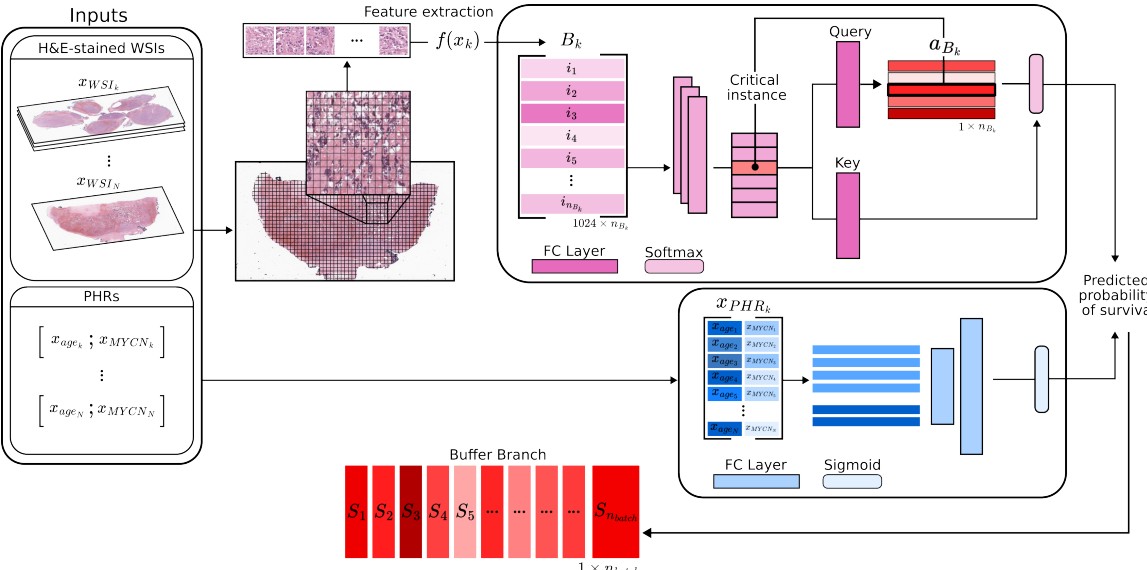

Figure 1: Infographic depicting the SurvivMIL pipeline. WSIs and patient-specific health records are considered in two separate branches and undergo unique preprocessing and predictive modelling. The branch predictions are combined in the projection head and stored in a buffer branch for an iterative concordance-based loss function calculation.

For the PHR branch, to transform the sparse clinical information into feature vectors suitable for efficient use by neural networks, we utilise a multi-layered perceptron (MLP). This transformation enhances the compatibility of the data with neural network architectures, facilitating effective processing and analysis in our model.

## 2.3 Buffer Branch & Concordance Loss Formulation

Our multi-branch architecture incorporates two distinct loss components: a multi-branch loss that utilises the classification predictions from the model's projection head and a

concordance-based loss function derived from the predictions accumulated in the buffer branch.

**Multi-branch Loss:** The multi-branch loss function considers each of the modalities' respective time-bin classifications, as well as their joint modality prediction of survival probability. For the branch-specific classifications:

$$\mathcal{L}_{WSI} = \mathcal{L}_{CE}(\mathbf{B}_k, \mathbf{Y}) \tag{4}$$

$$\mathcal{L}_{PHR} = \mathcal{L}_{CE}(\mathbf{P}_k, \mathbf{Y}) \tag{5}$$

$$\mathcal{L}_{multi} = \frac{1}{2}\bigg(\mathcal{L}_{WSI} + \mathcal{L}_{PHR}\bigg), \tag{6}$$

where $\mathcal{L}_{WSI}$ and $\mathcal{L}_{PHR}$ are the cross-entropy loss functions specific to the WSI and patient-specific health records predictions.

**Buffer Branch:** To compute concordance for survival analysis, we need to accumulate predictions across patients. However, the implicit batch size limitation in traditional MIL survival analysis approaches poses a challenge. To overcome this, we introduce a buffer branch in our network architecture designed to accumulate predictions from the projection head of the network and iteratively compute a concordance-based loss function. Implementing a buffer branch allows the model to dynamically update its stored predictions during training, allowing for continuous refinement of the concordance-based loss estimation.

**Concordance Loss:** Utilising the accumulated predictions in the buffer branch, we define concordance based on pairs of samples $(k, j)$ that are comparable under certain conditions. Specifically, a pair of samples, $(k, j)$ are comparable if $t_k \neq t_j$ and $min(t_k, t_j) = 0$ (indicating that at least one event has occurred). A comparable pair is deemed concordant if $t_k < t_j$ and $\hat{S}_k > \hat{S}_j$. The concordance index is defined as:

$$C = \frac{\sum_{k \neq j} \mathbf{1}\{t_k < t_j\}\mathbf{1}\{\hat{S}_k > \hat{S}_j\}}{\sum_{k \neq j} \mathbf{1}\{t_k \neq t_j\}\mathbf{1}\{min(t_k, t_j) = 0\}} \tag{7}$$

$$\mathcal{L}_{C-index} = 1 - C, \tag{8}$$

where $\mathbf{1}$ is an indicator function and $\mathcal{L}_{C-index}$ is the concordance-based loss.

Combining the classification loss and concordance-based loss, our total loss is derived:

$$\mathcal{L}_{total} = \mathcal{L}_{multi} + \mathcal{L}_{C-index}. \tag{9}$$

## 3 Experimental Evaluation

### 3.1 Setup

We obtained a dataset comprising 189 high-risk pediatric neuroblastoma patients. Each patient record includes an unannotated H&E stained WSI as well as supplementary information regarding the age and MYCN status (MYCN amplified, or MYCN non-amplified) of each patient, which is derived from the Fluorescence In Situ Hybridisation (FISH) detection test (Misra et al., 1995). In this cohort, 30% of the patients are MYCN-amplified, and 68%

had a poor prognosis. For each patient, we collect WSIs with an average bag size of 5,483 $256 \times 256$ patches per image. To evaluate our model, we perform 5-fold cross-validation and report the concordance index (C-Index). Additionally, we compared the performance of SURVIVMIL using our implemented concordance-based loss (CI Loss) against the widely used negative log-likelihood loss (NLL Loss).

**Baselines:** We evaluate our methodology against a range of baselines, all utilising the same pre-trained feature extractor (Chen et al., 2024). The baselines considered include AMIL (Chen et al., 2022), an attention-based MIL framework; TransMIL (Shao et al., 2021), which employs patch-to-patch interactions using the Nyström method (Xiong et al., 2021); and DSMIL (Li et al., 2020). We also consider an existing multimodal baseline, MCAT (Chen et al., 2021), an early fusion genomic-guided co-attention architecture. To assess the performance of SURVIVMIL, we employ three different evaluation methods: (1) using WSIs as a single data modality, (2) using patient-specific health records as a single data modality, and (3) combining both data modalities. In the first approach, the set of feature representations obtained by the feature extractor are fed as an input to the models. For the second approach, a 3-layer MLP and a sparse neural network (SNN) (Jaume et al., 2024) are used to transform the sparse scalar representations of the patient records into feature vectors. Finally, for the multimodal baselines (3), except for the case of MCAT, we adapt the WSI classifiers by concatenating the two different data modalities and passing the concatenated feature vectors as inputs into the respective networks.

### 3.2 Results

Table 1 highlights the predictive performance of SURVIVMIL against baselines evaluated on WSIs and PHRs. SURVIVMIL outperforms both the unimodal and multimodal baselines, achieving $+0.5\%$ above AMIL in a unimodal setting, $+7.1\%$ compared to SNN, and $+3.2\%$ above AMIL in a multimodal setting. We propose that this performance is a result of its effective handling dimensionality-gapped multimodal integration and its employment of the concordance-based loss function.

**Early Fusion *vs.* Late Fusion:** Except for the case of TransMIL, we find that early fusion multimodal baselines, trained on the concatenated modalities, slump in predictive performance compared to when trained on WSIs alone. This highlights the challenge of using early fusion methodologies for multimodal data that exhibits large dimensionality gaps. In contrast, we demonstrate that SURVIVMIL, a late fusion alternative, improves performance in scenarios with extreme dimensionality gaps. Notably, our ar-

Table 1: Survival Prediction results on our in-house paediatric NB dataset. We report the average and standard deviation results of 5-fold cross-validation, highlighting the best performance in **bold**.

| | Method | C-index ($\uparrow$) |
|---|---|---|
| WSI | DSMIL | $0.637_{0.0392}$ |
| | TransMIL | $0.498_{0.0100}$ |
| | AMIL | $0.646_{0.0920}$ |
| PHR | MLP | $0.567_{0.0024}$ |
| | SNN | $0.580_{0.0074}$ |
| Multimodal | DSMIL | $0.618_{0.0506}$ |
| | TransMIL | $0.524_{0.0287}$ |
| | AMIL | $0.619_{0.0799}$ |
| | MCAT | $0.540_{0.0455}$ |
| | SurvivMIL (NLL Loss) | $0.634_{0.0428}$ |
| | **SurvivMIL (CI Loss)** | $\mathbf{0.651}_{0.0455}$ |

chitecture also outperformed MCAT, an early fusion architecture, further reinforcing the effectiveness of late fusion in handling extreme dimensionality-gapped multimodal data.

**CI Loss *vs.* NLL Loss:** While SURVIVMIL employed with the NLL loss function outperforms the other multimodal baselines, it is the addition of the buffer branch, facilitating the CI Loss function of our network that provides the best overall performance across all unimodal and multimodal baselines. This combination effectively handles small batch sizes by accumulating predictions and directly optimising for concordance, resulting in more accurate survival predictions and better model calibration for our SURVIVMIL pipeline.

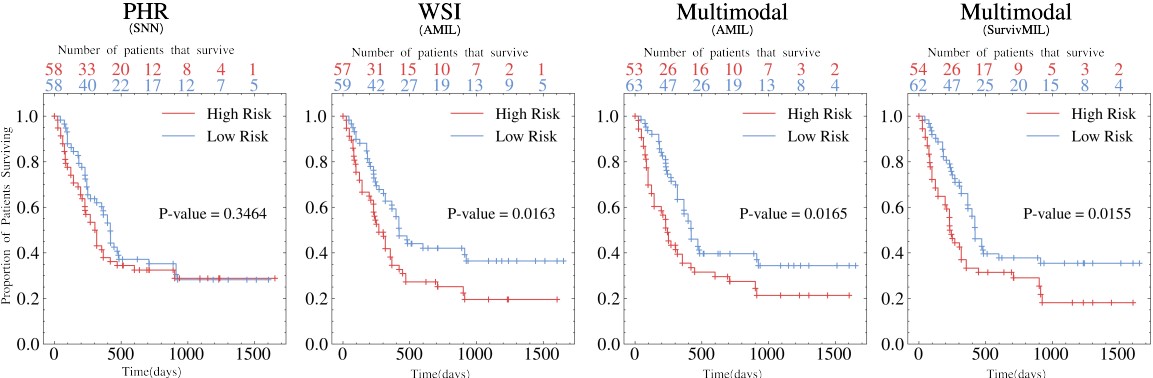

Figure 2: Kaplan-Meier curves for SURVIVMIL, compared with the best-performing models in both unimodal (PHR and WSI) and multimodal settings. Risk groups are defined using the mean of the survival probability predictions. The log-rank test was employed to evaluate statistical significance ($\alpha = 0.05$).

**Kaplan-Meier Analysis:** Figure 2 displays the Kaplan-Meier survival curves for high- (red) and low- (blue) risk groups predicted by the models. By statistical significance, SURVIVMIL achieves better delineation between the two risk groups compared to the best-performing unimodal and multimodal baselines. This improved delineation can be attributed to the late fusion architecture, which allows the network to best integrate and leverage the strengths of the two modalities towards a more refined prediction of a patient's risk. Additionally, within our cohort of high-risk neuroblastoma patients, we demonstrate that morphological differences across diverse patient samples can be leveraged to successfully sub-stratify an 'ultra-high-risk' patient group.

**Impact of Modality Contributions on Predictive Performance:** In Figure 3, we assess the impact of varying the contributions of different modalities within our SURVIVMIL architecture on predictive performance. Our findings indicate that model performance improves as more diverse modalities are integrated. The optimal combination for overall survival prediction is achieved when whole-slide images (WSIs) contribute a weighted proportion of 0.3 and patient-specific health records contribute 0.7.

## 4 Conclusion

This paper focuses on neuroblastoma (NB) patient outcomes and addresses two key challenges in multimodal survival analysis using Multiple Instance Learning (MIL):

**Exteme Dimensionality-Gapped Multimodal Integration:** The first challenge involves defining architectures that effectively integrate diverse modalities, especially when there are extreme data dimensionality gaps. We propose a late fusion architecture that allows for weighted attribution of each modality, enhancing the accuracy of NB patient outcome predictions.

**MIL Framework Limitations:** The second challenge is overcoming the limitations posed by the MIL framework in survival analysis, where the hazard function is typically approximated using a parameterised log-likelihood loss function due to batch size constraints. We address this by incorporating a buffer branch into our architecture, designed to accumulate a minibatch of predictions, enabling the use of an efficient concordance-based loss function. Our model demonstrates improved performance over all unimodal baselines and surpasses all early fusion multimodal baselines. Additionally, we show that our model achieves better predictive performance when employing the concordance index (CI) loss function compared to the negative log-likelihood (NLL) loss function.

In the context of our high-risk NB patient cohort, we present a successful, statistically significant sub-stratification of a proportion of high-risk patients into an ultra-high-risk group. Further research into better understanding the morphological biomarkers specific to this subgroup of categorised patients could facilitate enhanced precision in diagnosis at an earlier stage.

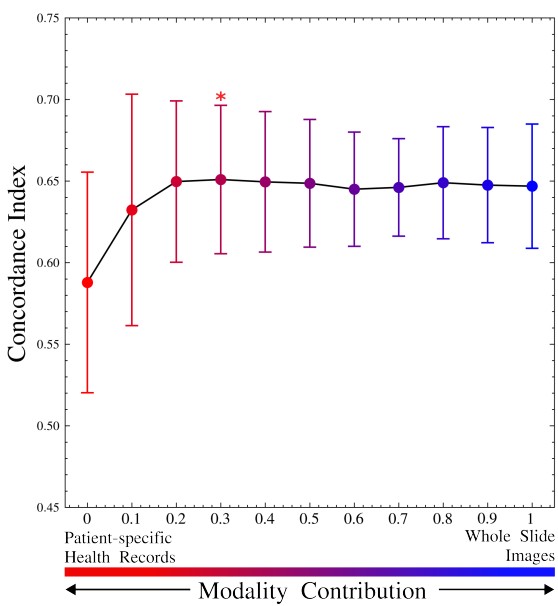

Figure 3: Curve depicting the change in concordance index performance for SurvivMIL as different weighted contributions from either modality attend to the prediction of patient outcomes.

## Acknowledgments and Disclosure of Funding

This is a summary of independent research supported by the National Institute for Health Research (NIHR) Biomedical Research Centre at The Royal Marsden NHS Foundation Trust and The Institute of Cancer Research. The views expressed are those of the author(s) and not necessarily those of the NHS, the NIHR or the Department of Health and Social Care. The Authors would also like to thank StratMedPaediatrics2 funded by Cancer Research UK, CRCEMA-Jul23/100001

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
