# OpenReview forum: "SurvivMIL: A Multimodal, Multiple Instance Learning Pipeline for Survival Outcome of Neuroblastoma Patients"
_MICCAI.org/2024/Workshop/COMPAYL — COMPAYL 2024_

### Official Review · Reviewer_9fZ2 · 2024-07-03
**novel MIL pipeline for predicting survival in NB patients**

**Custom Rating:** 3
**Confidence:** 3

**Review:**

The authors proposed SurvivMIL, a model that integrates multimodal prognostic data for predicting NB patient outcomes. The model was evaluated on an in-house pediatric NB patient dataset.

Pros:
1. An interesting dataset is chosen for evaluation;
2. A late fusion architecture with weighted attribution of each modality was explored;

Cons:

My main concern is that the main contribution of the paper is in proposing the late fusion model that was tested on the in-house dataset only. From my perspective, it'd be crucial for acceptance of this paper either a) release the dataset to the public, or if not possible then b) test the approach on publicly available data (TCGA?).

Another potential way to make the paper stronger and more beneficial for the community could be to release a python tool / library that can be easily reusable so others can easily use the tool and benefit from the proposed approach.

Finally, it's not that clear to me from the paper how the proposed approach is different from other approaches?

There are some other relevant overviews on different multi-modal fusion strategies that you could mention:

1. https://www.ncbi.nlm.nih.gov/pmc/articles/PMC10484010/
2. https://arxiv.org/html/2303.06471v3

---

### Official Review · Reviewer_dcPw · 2024-07-10
**The study is well-motivated and well-designed, and the manuscript is well-written. However, there are parts, mainly in the methodology, that could have been better explained.**

**Custom Rating:** 4
**Confidence:** 5

**Review:**

The authors describe a multi-branch pipeline for survival analysis of high-risk neuroblastoma cancer patients with a late fusion approach of H&E WSIs and patient-specific health records (PHRs). The study also addresses the challenge of integrating modalities with different dimensionalities, since PHRs contain sparse information and WSIs are highly information-dense and processed at high resolution. Additionally, a buffer branch was introduced to address the batch size limitation in traditional MIL survival analysis approaches.

Comments regarding the manuscript and the application:
- The dataset is not well described. It is stated that 189 high-risk patients are utilized, but no other risk-ranking cases are mentioned. Also, the train, validation, and splitting partitions are not mentioned.
- The multi-branch classifier is not well described and depicted in Figure 1:
   - It is mentioned that the instance-level classifier performs a max-pooling operation to determine a critical instance. The naming instance-level classifier is confusing since no classification downstream is performed at this part of the algorithm, but the max-pooling operation is to select the most critical instance. In addition, max-pooling is a predefined and non-trainable operation and thus reduces the flexibility of a MIL model [1]. A solution to this issue is to use attention-pooling, which is a fully flexible MIL pooling that can be trained alongside other components of a model. Furthermore, with max-pooling, probably the algorithm will always select the same instance of a WSI bag as a critical instance in every epoch and fold.
  - A softmax layer is used in the WSI branch, and the definition of the denominator of the softmax function is the sum of the exponentials of all the elements in the input vector over the possible outcomes, where the outcomes are the classes. However, there are no classes clearly described in the paper. Are the classes the risk ranking of the patients?
  - It is stated that a late fusion approach in the multi-branch classifier was utilized, combining the global outputs from the WSI data with those from PHRs. However, it is unclear how the late fusion was performed, how the predictions of the branches were combined, and how the projection head is structured.
  - In Figure 1, an attention matrix is depicted, but the critical instance probably has a dimension of $1\times1024$ multiplied by     $(N\times1024)^T$, and this multiplication outputs a $1 \times N$ value and not a matrix.
- The C-indices of the unimodal AMIL and multimodal AMIL are very close, and a statistical comparison would tell if there is a statistically significant difference between the performance of those models.
- In the Kaple-Meier analysis, a statistical comparison is performed between 4 models at statistical significance value a=0.05, and it is stated that SURVIVMIL achieves significantly better delineation. However, the Bonferroni correction would be suitable since four statistical comparisons on the same data are performed.

[1] Ilse, Maximilian, Jakub M. Tomczak, and Max Welling. "Deep multiple instance learning for digital histopathology." Handbook of Medical Image Computing and Computer Assisted Intervention. Academic Press, 2020. 521-546.

---

### Official Review · Reviewer_TnA6 · 2024-07-12
**multimodal approach with WSIs and PHR.**

**Custom Rating:** 3
**Confidence:** 4

**Review:**

**Overview**

This paper explores a multimodal approach combining imaging data from Whole Slide Images (WSIs) with patient-specific health records (PHR). While the former comprises high amounts of data, thus requiring multiple-instance learning (MIL), the latter is much more condensed. The authors propose a late fusion approach for WSI and PHR data. Furthermore, a new loss function with a buffer branch is proposed to mitigate the problems of small batch sizes, while being fit to ranking survival. The method is applied to the survival analysis of neuroblastoma cancer patients in a private dataset. While the method contains some novelty in data usage, the results and improvements are unclear.

**Pros**

1. The text is well-written and easy to follow.
2. The new loss contains some novelty.
3. The motivation for the concordance-based loss function is sound in exploring the ranking of survival predictions.

**Cons**

1. While the usage of these data in this specific application, late fusion has been explored before [1].
2. The dataset is relatively small (n=189) and in-house, thus, making it hard to reproduce results or compare against.
3. The results do not fully show the benefits of the proposed method. The benefits of multi-modality are unclear.


**Further comments**

Regarding the results, in future research, the work could be strengthened with stronger results, if possible. Perhaps, by experimenting on different datasets, too.

    1. In Table 1, multimodal DSMIL and AMIL are significantly worse than their unimodal counterparts. The authors state that this is for the early fusion approach. However, one could expect that the new data would be just ignored due to the lower dimension, but the gap to unimodal is large. Could the authors elaborate further on why multimodal leads to a performance decrease, please? Does the multi-modality setting bring benefit, indeed?

    2. The authors claim that multimodal late fusion (SurvivMIL NLL Loss, Table 1) is effective. However, when we compare that model to the best of WSI-only models, we observe that the results are on par. Therefore, would the proposed WSI-only MIL method adopted here underperform without extra modalities?

    3. In Fig. 3, after contribution > 0.2, the results do not vary too much. The figure is interesting in showing that WSI-based data helps over PHR. Also, it shows that the model becomes more reliable, as the confidence bars decrease. Still, it is hard to see a clear benefit of the proposed method


**References**

[1] Chen, Richard J., et al. "Pathomic fusion: an integrated framework for fusing histopathology and genomic features for cancer diagnosis and prognosis." IEEE Transactions on Medical Imaging 41.4 (2020): 757-770.

---

### Decision · Program_Chairs · 2024-07-16

Accept